# Using Deep Reinforcement Learning to Train and Evaluate Instructional Sequencing Policies for an Intelligent Tutoring System

## Abstract

We present STEP, a novel Deep Reinforcement Learning solution to the problem of learning instructional sequencing. STEP has three components: 1. Simulate the tutor by specifying what to sequence and the student by fitting a knowledge tracing model to data logged by an intelligent tutoring system. 2. Train instructional sequencing policies by using Proximal Policy Optimization. 3. Evaluate the learned instructional policies by estimating their local and global impact on learning gains. STEP leverages the student model by representing the student's knowledge state as a probability vector of knowing each skill and using the student's estimated learning gains as its reward function to evaluate candidate policies. A learned policy represents a mapping from each state to an action that maximizes the reward, i.e. the upward distance to the next state in the multi-dimensional space. We use STEP to discover and evaluate potential improvements to a literacy and numeracy tutor used by hundreds of children in Tanzania.

## 1 Introduction

An Intelligent Tutoring System (ITS) aims at teaching a set of skills to users by individualizing instructions. Giving instruction to users requires many sequential decisions, such as what to teach, what activities to present, what problems to include, and what help to give. Our aim is to take decisions which maximize long-term rewards in the form of learning gains, so Reinforcement Learning (RL) is a natural approach to pursue, and was first proposed by Liu (1960).

The goal of an RL agent is to learn a policy $\pi$, defined as a mapping from state space $S$ to action space $A$. Given any state, the RL agent follows a series of actions proposed by the learned policy to maximize the long-term expected reward. In the context of an ITS, we specify the RL agent as follows:

- State $s_t$: We define the state as a combination of the student state and the tutor state. The tutor state determines the set of actions available to the RL agent at a given timestep. We represent the student state as a vector of probabilities where element $i$ is the estimated probability that the student knows skill $i$.

- Action $a_t$: The action taken by the RL agent corresponds to a tutor decision at a particular grain size.

- Reward $r_t(s_t, a_t)$: Defined as the average difference between prior and posterior knowledge states based on the simulated student's response to the tutor action $a_t$ to the student simulator.

- Next state $s_{t+1}$: The knowledge vector of a student after a Bayesian update based on the simulated student's response to tutor action $a_t$ in state $s_t$ is the updated student knowledge state. The updated tutor state is given by the tutor simulator. The updated student knowledge state and tutor state, together gives the next state $s_{t+1}$.

We instantiate STEP in the context of RoboTutor, a Finalist in the Global Learning XPRIZE Competition to develop an open source Android tablet tutor to teach basic literacy and numeracy to chil-

dren without requiring adult intervention. XPRIZE independently field-tested the Swahili version of RoboTutor for 15 months in 28 villages in Tanzania.

Figure 1 shows an diagrammatic overview of STEP and the rest of the paper is organized as follows. Section 2 discusses the simulation of tutor and student (the environment block). Section 3 elaborates on the training of decision policies (the RL agent block). Section 4 evaluates the learned policies. Section 5 relates this work to prior research. Section 6 concludes.

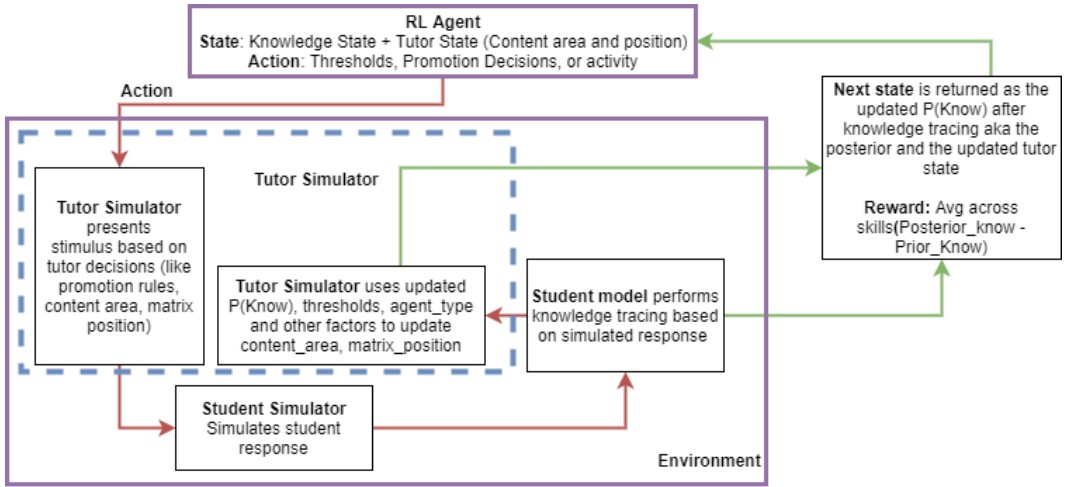

Figure 1: The RL setup for STEP

## 2 SIMULATING THE TUTOR AND THE STUDENT

To apply RL, we need to simulate the tutor's actions and the student's responses to them.

### 2.1 TUTOR SIMULATOR

The data for this paper comes from the version of RoboTutor used during the last 3 months of XPRIZE's 15-month field study. This version rotates through three content areas (literacy, numeracy, and stories), tracking the child's position in each area's curricular sequence of successively more advanced activities. It lets the child select among doing the activity at that position, advancing to the next activity, repeating the same activity (from the previous content area), or exiting RoboTutor. After selecting an activity, the child may complete all or part of it before selecting the next activity. RoboTutor has 1710 learning activities, each of which gives assisted practice of one or more skills on a sequence of items, such as letters or words to write, number problems to solve, or sentences to read. Each item requires one or more steps. Each step may take one or more attempts.

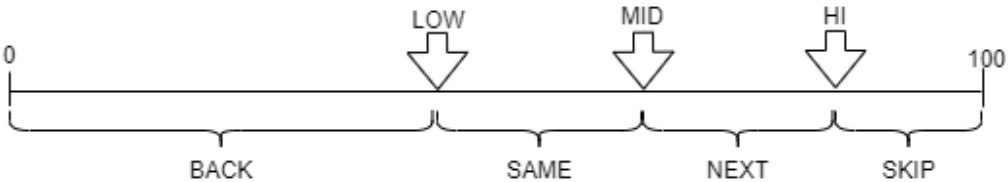

Figure 2: Threshold on percentage of correct attempts and their effects on tutor decisions.

The simulated tutor state identifies the current content area and the child's position in it. RoboTutor (actual or simulated) updates the position in the content area based on the percentage of correct attempts to perform the steps in an activity. Specifically, it uses fixed heuristic thresholds (called

LOW, MID, HI) on this percentage to demote BACK to the previous position, stay at the SAME position, promote to the NEXT position, or SKIP to the position thereafter. Figure 2 gives an illustration of the same.

## 2.2 STUDENT SIMULATOR

A student simulator should behave like students who use the tutor. Accordingly, the simulator uses a Bayesian Knowledge Tracing (BKT) student model trained on logged data using HOT-DINA. It has the same Guess, Slip, and Learn parameters as standard BKT, but estimates the Knew parameter based on skill difficulty and discrimination and student proficiency from Item Response Theory. Thus, HOT-DINA extrapolates from the student's knowledge of other skills, and other students' knowledge of this skill, albeit at a high computational cost to fit the model. Xu & Mostow (2014) found HOT-DINA to have higher predictive accuracy than standard BKT.

To limit computation time, we fit the model on logged data from a single village, consisting of 42,010 attempts by 8 children to apply 22 skills. We fit one proficiency parameter for each child and 5 parameters for each skill (Guess, Slip, Learn, Difficulty, and Discrimination), 118 parameters in total. (Fitting 5 separate parameters per activity instead of per skill might achieve higher accuracy but would require fitting 8,558 parameters.) We use MCMC sampling for Bayesian inference with PyStan rather than the OpenBUGS Gibbs sampling package used in the original HOT-DINA work because PyStan is faster and handles larger datasets. Nevertheless, fitting the 118-parameter HOT-DINA model to 42,010 attempts took approximately 4 days on a supercomputer with 128 GB and 28 cores.

Table 1: Converged parameters for the HOT-DINA student model

| Parameter | Mean (converged) | SD (converged) |
|-----------|------------------|----------------|
| $\theta_1$ | -0.17 | 0.93 |
| $\theta_2$ | 1.77 | 0.64 |
| $\theta_3$ | -0.65 | 0.72 |
| $\theta_4$ | -0.73 | 1.13 |
| $\theta_5$ | -0.38 | 0.78 |
| $\theta_6$ | -0.28 | 0.55 |
| $\theta_7$ | -0.48 | 0.7 |
| $\theta_8$ | 0.14 | 0.47 |
| $b_1$ | 1.25 | 0.73 |
| $b_2$ | 1.25 | 0.72 |
| $b_3$ | 1.25 | 0.72 |
| $b_4$ | 1.25 | 0.72 |
| $b_5$ | 1.42 | 0.71 |
| $b_6$ | 1.49 | 0.68 |

Table 1 shows converged values for a subset of HOT-DINA parameters. For example, the eight $\theta$ values refer to the 8 student proficiency parameters of the student model. For simplicity, we show only the first 6 values of $b$ (skill difficulty parameter) in the table. Once we obtain the model parameters, we need two things to be done for the student simulator to be successful: given an activity, we should be able to simulate whether a student gets an activity right or wrong. Based on this response, we should be able to perform knowledge tracing over multiple skills to update the student's knowledge probabilities. For simulating a student's performance on an activity we first estimate P(Getting Activity $j$ Correct) as in equations 2 and 3 below. We then simulate the student response (right or wrong) by doing a biased coin flip based on this estimated probability. Since we now have a simulated student response, we perform knowledge tracing over multiple skills using the update equations 3-5. The next few lines cover some basic notation and update equations for simulated learning of a student. It should be noted that variables $\alpha, y, \text{ and } Y$ are all binary, ie., they take on value of either 1 or 0.

$\theta_n$        Proficiency of student n
$a_k$        Discrimination of skill k

$b_k$            Difficulty of skill k

$q_{jk}$          1 if activity j exercises skill k, 0 otherwise

$\alpha_{nk}^{(t)} = 1$     Probability that student n knows skill k at time-step t

$y_{nk}^{(t)} = 1$     Probability that student n answers an activity exercising only skill k at time-step t

$Y_{nj}^{(t)} = 1$     Probability that student n gets activity j correct at time-step t

$$\alpha_{nk}^{(0)} = \prod_{k=1}^{K} \left( \frac{1}{1 + exp(-1.7 a_k(\theta_n - b_k))} \right)^{q_{jk}} \tag{1}$$

$$(y_{nk}^{(t)} = 1) = (1 - slip_k)(\alpha_{nk}^{(t)} = 1) + guess_k(\alpha_{nk}^{(t)} = 0) \tag{2}$$

$$(Y_{nj}^{(t)} = 1) = \prod_{k=1}^{K} (y_{nk}^{(t)} = 1)^{q_{jk}} \tag{3}$$

$$(\alpha_{nk}^{(t)} = 1 | Y_{nj}^{(t)} = 1) = (\alpha_{nk}^{(t)} = 1) * \left( \frac{1 - slip_k}{(y_{nk}^{(t)} = 1)} \right)^{q_{jk}} \tag{4}$$

$$(\alpha_{nk}^{(t)} = 1 | Y_{nj}^{(t)} = 0) = (\alpha_{nk}^{(t)} = 0) * \left( \frac{guess_k}{(y_{nk}^{(t)} = 1)} \right)^{q_{jk}} \tag{5}$$

$$(\alpha_{nk}^{(t+1)} = 1) = (\alpha_{nk}^{(t)} = 1 | Y_{nj}^{(t)}) + (learn_k * (\alpha_{nk}^{(t)} = 0 | Y_{nj}^{(t)})) \tag{6}$$

## 3   Training policies with PPO

We have already discussed the student simulator and tutor simulator in last section. In this section, we discuss the training a policy using STEP in the context of RoboTutor.

### 3.1   The reward function

The RL agent learns a decision policy – that is, a mapping from states to actions – that maximizes the total expected reward of following the policy $\pi_\theta$. As the reward function for student n, we use the knowledge gain as estimated by the student model, i.e. posterior minus prior estimates of Pr(student i knows skill k), averaged over all skills. The posterior and prior refer to the knowledge states before and after applying the bayesian updates (equations 3-5) on an activity decided by action $a_t$. The information for prior knowledge is implicitly present in the knowledge state of $s_t$ In order to save computational time, we learn policy for episodes of 100 timesteps using PPO after which the episode terminates. Though our experiments stick to finite-horizon undiscounted returns with 100 steps, it is trivial to extend this approach to any finite number of steps or even to infinite-horizon discounted returns with discount factor $\gamma \in (0, 1)$ so the rewards vanish at large timesteps. The reward function $r_t$ for student n at a given step is given by learning gains of a student due to attempting an activity, as given in equation (7) where K is the total number of skills (22 for RoboTutor). The returns are just the sum of rewards over $T$=100 steps. (Previous methods used reward=0 or 1 based on correct attempt or something else. Useful to mention this?)

$$r_t(s_t, a_t) = \frac{\sum_{k=1}^{K} (\alpha_{nk}^{(t+1)} = 1) - (\alpha_{nk}^{(t)} = 1)}{K} \tag{7}$$

According to the student model trained by HOT-DINA on the 8 children's log data, their prior averaged 0.55 and their posterior averaged 0.73, a gain of 0.18 over their final usage consisting of 42,010 attempts (up to 3 months). Their posterior after their first 100 attempts averaged across the 8 students was 0.64, for an average gain per attempt of 0.09/100 = 0.0009.

We can train different types of RL agents depending on their state space and range of actions, which depend on how far they depart from RoboTutor's current decision policy.

### 3.2 STATES, ACTIONS AND RL AGENT TYPES

We model student n's state as the vector of estimated probabilities of student n knowing skill k $[(\alpha_{n1}^{(t)} = 1), ..., (\alpha_{nK}^{(t)} = 1)]^T$. Depending on the RL agent type, the tutor state may include the current (active) content area (literacy, numeracy or stories) and the student's current position in the curricular sequence for that area; just the content area; or neither.

Alternative ranges of actions for each agent type:

- Type 1: 3 threshold actions (LOW, MID, HI), each action $\in (0.0, 1.0)$
- Type 2: promote-demote decisions choosing one of BACK, SAME, NEXT, and SKIP. 1 action from a Discrete(4) action space.
- Type 3: an activity from the current content area. 1 action from a Discrete(x) action space where x is the number of activities in the current content area.
- Type 4: any activity from any content area. 1 action from a Discrete(1710) action space.

Table 2 summarizes 4 types of RL agents we consider, whose tutor simulators operate in the following ways:

- Type 1 preserves RoboTutor's current choices but adjusts thresholds that affect promote-demote decisions indirectly
- Type 2 eliminates the need for thresholds by choosing promote-demote decisions directly from state rather than from thresholds
- Type 3 can jump to any activity within the current content area
- Type 4 can jump to any activity in any content area; area rotation constraint is removed

Table 2: Tutor state and action range for each agent type

| Type | Tutor State | Action Range | #Actions |
|------|-------------|--------------|----------|
| 1 | Content area, position in curriculum | Thresholds | 3, Continuous |
| 2 | Content area, position in curriculum | BACK, SAME, NEXT, SKIP | 1, Discrete |
| 3 | Content area | Any activity in content area | 1, Discrete |
| 4 | None | Any activity in any content area | 1, Discrete |

## 4 EVALUATING LEARNED POLICIES

We evaluate the learned policies along two metrics which assess the local and global impacts. The local impact is the average change in reward by replacing a single historical choice of activity with the activity chosen by the policy. The global impact is the overall change in reward per attempt by following the learned policy from the first attempt. Table 3 evaluates the learned policies, per agent type, by their impact on learning gains (expected reward) of the first 100 attempts averaged across the 8 children, compared to the historical baseline of 0.0009.

Table 3: Evaluating policies based on local and global impact

| Agent Type | Local impact | Global impact |
|------------|--------------|---------------|
| 1 | 9.98x | 1.37x |
| 2 | 10.81x | 1.44x |
| 3 | 10.99x | 1.66x |
| 4 | 11.64x | 2.47x |

From the table, we see an increasing trend for both the local and global impacts and is in-line with our beliefs that less constrained the RL agent is, the greater the impact. Interestingly, local impacts seem to *exceed* global impacts for all 4 cases. This is because successive attempts have independent

rewards since the Prior(Know at step $t$) does not depend on the policy-proposed-action at step $t-1$ for local impacts. Thus, if some less-known activity allows a large one-time gain at step $t-1$, the local impact allows it to occur multiple times in subsequent steps, beating average global gains per step.

Figure 3 evaluates the current RoboTutor policy (red) versus the agent's learned policy on the simulated student for agent types 1 to 4. Every agent type has 8 subplots associated to each simulated student built off the 8 students' data that we logged through RoboTutor. The y-axis corresponds to student's average knowledge across skills and the x-axis corresponds to the number of attempts of a student. Since we restricted the time-horizon to 100, we can see that the x-values have an upper limit of 100 attempts.

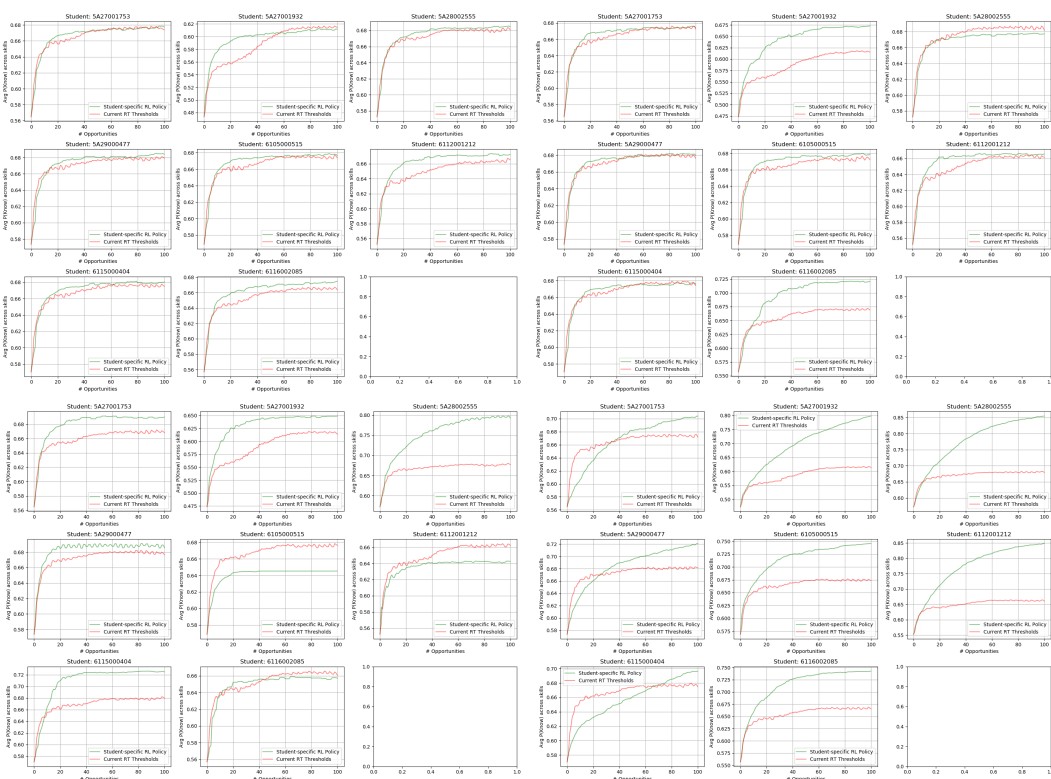

Figure 3: RoboTutor policy (red) vs learned policy (green) studying student knowledge vs attempts for all 4 agent types

## 5 RELATION TO PRIOR WORK

Various researchers have worked on Reinforcement Learning for instructional sequencing Doroudi et al. (2019). Table 4 summarizes work that used BKT for student modeling or Deep RL for optimization.

Prior work by Yudelson et al. (2013) and Pardos & Heffernan (2011) used BKT methods that fit a parameter for the probability of already knowing a skill prior to instruction. In contrast, we use HOT-DINA, a higher-order BKT-IRT hybrid that estimates this probability based on skill difficulty and student proficiency, achieving higher accuracy than standard BKT.

Recent work by Shen et al. (2018) on instructional sequencing used deep reinforcement learning, specifically Deep Q-Networks. STEP uses a more powerful deep RL method, namely *Proximal Policy Optimization (PPO)* Schulman et al. (2017).

Some prior work reviewed by Doroudi et al. (2019) specifies the reward as 1 when the probability of knowing a skill reaches 0.95 and 0 otherwise. In contrast, we define reward as estimated learning gain, so as to differentiate between actions that yield different gains in student knowledge.

Table 4: Prior work and their approaches

| Paper | Student model | Adaptive policy |
|---|---|---|
| David et al. (2016) | BKT | Threshold |
| Whitehill & Movellan (2017) | POMDP | Policy Gradient |
| Doroudi et al. (2017) | BKT | Mastery |
| Shen et al. (2018) | Model-Free | DQN |
| Segal et al. (2018) | BKT | Threshold |
| Doroudi et al. (2019) | BKT | Inc Time |
| **This paper** | **HOT-DINA** | **PPO** |

## 6 CONCLUSION

This paper contributes a novel framework for optimizing instructional sequencing in an intelligent tutoring system by combining knowledge tracing with deep reinforcement learning and evaluating the learned decision policy on historical data. We fit a simulated student on authentic log data from real children using RoboTutor in Tanzania, in contrast to earlier work that used synthetic data. We trained the student model using HOT-DINA because it is more accurate than other knowledge tracing methods. We used Proximal Policy Optimization because it learns better than previous reinforcement learning methods applied previously to ITS. We use knowledge probabilities estimated by the student model as a state and directly optimize for learning gains which we set as the reward. We evaluated the learned policies' local and global impact on expected knowledge gains relative to a historical baseline and explained the somewhat surprising results we observed.

The work has several limitations. The evaluation is based on data from 8 children from one village to save computational expense. The evaluation extrapolates from historical data. Future work should test the actual impact of learned policies on children's learning. We also do not make predictions on kids backing out while doing activities and remove the 10-item per activity constraint while performing our experiments. Future work should include predicting student disengagement.

We use a 118-parameter HOT-DINA model to save computational expense, while the 8,558-parameter HOT-DINA model might have been more accurate since it has parameters per activity instead of per skill. Developing other student models that are more accurate than HOT-DINA might be fruitful. We focused on learning decision policies for choosing activities. Future work could explore optimizing tutor decisions at other levels of granularity, such as selecting which items to practice and what assistance to provide.

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
