# OpenReview forum: "Using Deep Reinforcement Learning to Train and Evaluate Instructional Sequencing Policies for an Intelligent Tutoring System"
_ICLR.cc/2021/Conference — Reject_

### Official Review · AnonReviewer3 · 2020-10-25
**Needs improvement in empirical rigour, novelty, clarity, and reproducibility. Recommend rejection.**

**Rating:** 2
**Confidence:** 3

**Review:**

	1. Summary of paper:
		a. The paper contributes a deep RL approach to learning instructional sequencing. The approach called STEP starts by simulating tutor and student models. The tutor simulation is based on the RoboTutor ITS and the student simulations are fit to historical data of children interacting with RoboTutor using the HOT-DINA approach. The instruction sequencing model is then trained using PPO and evaluated using novel measures of sequencing decision impact on local and global learning gains (estimated from running student simulation). The paper contributes one set of experimental comparisons between four variations of the sequencing agent and the RoboTutor baseline in 8 runs of the two systems.
	2. Strengths
		a. At a high level, the proposed approach is well motivated, using RL to optimise parameters in the existing tutoring system or more granularly make sequential decisions about what activity to provide to the student.
		b. The local and global reward design is a strong contribution that uses historical data and the student simulation (knowledge tracing) to estimate credit for policy decisions in a counterfactual manner.
		c. The future work discusses the limitation of not using actual children's scores to evaluate this learning model.
	3. Weaknesses
		a. Novelty/impact and context within related literature:
			i. It is difficult to judge the novelty of this work since the related work section is too brief and does not actually describe several works compared against. Additionally, table 4 is not descriptive enough, has potentially relevant work undescribed (Whitehall & Movellan's 2017 POMDP/policy gradient approach), and has work referenced previously missing from it (Yudelson et al. 2013, Pardos & Heffernan 2011).
		b. Experimental rigour:
			i. Two strong claims are made in the related work section that do not have sufficient experimental evidence. 1) A direct comparison of BKT to HOT-DINA for modelling student knowledge gains is required to show the impact of this central claimed contribution. 2) A direct comparison against Shen et al. 2018 is required to show that PPO is indeed more effective in this domain. This is also important since it is a central claimed contribution.
		c. Clarity: The clarity of the paper needs significant effort to improve. Instances below.
			i. The structure of the paper reads like a technical report rather than an empirical investigation. The contents would be far easier to understand with a different structure emphasising a research question, background to understand/motivate it, methodology to answer it, results, and discussion.
			ii. Several sections are far clearer in the supplementary data document. The reproducibility of the paper is boosted by this. Given the extra space available, several sections could stand to be transferred to the main paper. My original review did not include supplementary data and points around describing data and examples are boosted by adding them to the main paper.
			iii. Until significant rereads, it isn't clear what the relationship is between RoboTutor and STEP/this work. With my current understanding, RoboTutor is an external system that has been used to collect data on children learning various skills using it. This data was then used to evaluate STEP in terms of estimated learning gains.
			iv. The tutor simulator section describes how RoboTutor functions. The student simulator describes how the knowledge tracing model works. Example differences between activities, skills, steps, etc. would make the content clearer.
			v. In the tutor simulation section it states that the child can select activities, so this part is replaced by RL decision-making in agent type 3 and 4, right? This relates to my question about RoboTutor and this work. I am understanding that the current work simulates the exact decision-making process of RoboTutor but can vary/change that process according to the different agent types. Is that correct?
			vi. Bayesian Knowledge Tracing needs a citation and a brief explanation in a background section.
			vii. HOT-DINA needs a citation and a brief explanation in a background section.
			viii. Item Response Theory needs a citation and a brief explanation in a background section.
			ix. What is the difference in computational cost between a BKT approach and HOT-DINA?
			x. A clearer highlighting of what data was used would make it easier to read the article. What was the contents of the data collected to enable knowledge tracing in the student simulator? This also ties in to the comment about examples. Adding a running example of an activity, skill, step for the tutoring task would enable a description of what data is collected to measure student knowledge in the data set.
			xi. What do the guess, slip, learn, etc. Parameters measure?
			xii. "We use MCMC sampling for Bayesian inference with PyStan rather than the OpenBUGS Gibbs sampling package used in the original HOT-DINA work because PyStan is faster and handles larger datasets." This line needs references for all software used, but most importantly, a reference to the original HOT-DINA work is necessary.
			xiii. The sole paragraph on page 3 is difficult to parse and seems important to understand how the student simulator works. The paragraph is dense and conversational in style referencing a sequence of steps without making them clearer and referencing equations that are on the next page. It would be much clearer to have the exposition and equations interspersed and use pseudo-code or a flowchart to explain the steps performed to simulate the student (at least a list).
			xiv. "We can train different types of RL agents depending on their state space and range of actions, which depend on how far they depart from RoboTutor’s current decision policy." This would make sense as the start of a new section on the experimentation.
			xv. A stronger partitioning of content would also be achieved by calling a potential new section at this point methodology, experiments, or evaluation. This would also help organise the next section of state, action, and agent types into a concrete experiment for which the paper is describing the state and action spaces.
			xvi. In section 3.2, it is confusing to have states, then actions, then agent types described. It seems like one experiment with 4 experimental variants (agent types) and a baseline (RoboTutor).
			xvii. Table 2 does not convey much more information than the explanation before it.
			xviii. Figure 4 is very difficult to parse. Instead of showing 36 subplots (with 4 empty subplots) for 8 students and 4 agents comparing against the RoboTutor baseline, it would be far easier to compare performance against students or against agent types, by combining all 8 student type runs for each variant into a single variance-shaded run in a single graph (e.g. using https://matplotlib.org/3.1.1/api/_as_gen/matplotlib.pyplot.fill_between.html). That would allow for much higher information density and an at a glance comparison between the 5 runs being compared combined across all 8 students. At the very least this should be done by combining all 5 runs into 8 separate graphs, though the previous approach is preferable.
			xix. Related work is far too brief and does not make clear what is being compared for many cited works. E.g. It isn't clear why the works that specify reward a certain way in Doroudi et al. (2019) show a disadvantage to the current approach, the works in table 4 don't specify why the current approach is an advance over their contributions.
		d. Reproducibility:
			i. Section 3 (and the paper in general) contains far too little information about the policy representation, learning hyperparameters, network architecture, etc. to understand the contribution.
	4. Recommendation:
		a. Per the weaknesses in the review above, I recommend the paper for rejection. I don't think the weaknesses in experimental rigour can be fixed in time, content space, or degree to support acceptance. The significant editing required to fix the other issues also seem unrealistic in time and space.
	5. Minor Issues
		a. "(Previous methods used reward=0 or 1 based on correct attempt or something else. Useful to mention this?)" There is a comment remaining in the paper that should have been removed.

---

> ### Author Response · Authors · 2020-11-13
> **Thanks for helpful reviews**
>
> AnonReviewer3 - thanks a lot for your valuable reviews/feedback. We shall keep these comments in mind as we try to improve our work in the future.

---

### Official Review · AnonReviewer2 · 2020-10-27
**Good idea and the results are showing promise, but the paper is not ready for publication in its current form.**

**Rating:** 4
**Confidence:** 4

**Review:**

Summary:
The paper describes variants of an intelligent tutoring system (ITS) developed using a newer (but previously published) variant of Knowledge Tracing (HOT-DINA) for assessing student proficiency and an RL algorithm (PPO) for making decisions on items and content areas to try next.  An empirical simulation calibrated to 8 students is reassessed on the same student simulations and improvements over the original tutoring system are empirically demonstrated.  Four variants with differing levels action granularity and knowledge racing are analyzed.

Review:
I really like the direction this paper is going in but the results as currently reported seem rushed and under-analyzed.  The paper is shorter than the max length of ICLR papers yet omits crucial details about the action selection strategies and presents the empirical gains with charts that lack labels and no textual analysis (just graphs without context) of the individual student trajectories.  There is also no comparison to the existing state-of-the-art from Shen et al. (cited).  Also, the empirical study seems to have been done on the same 8 students on which the model was calibrated, which likely caused significant overfitting and puts the generality of the empirical results in doubt. Finally, much of the IRT terminology is not defined until later in the paper and the related work on RL for intelligent tutoring systems could use some additions (mentioned below).  In summary, this is a good idea and the results are showing promise, but the paper is not ready for publication in its current form.

Details:
My biggest concern with the paper is the lack of rigor in the empirical analysis.  First and foremost, it appears the data from the same 8 students were used for calibrating the model (both the knowledge tracing and the PPO decision-making training) and then those same students were considered in the simulation (testing) of the models.  It seems likely the models were overfit to the data from these students.  A proper empirical study in this kind of educational setting needs to train the models on data from one set of students and then use a holdout set to test it.

Next, the metrics and charts reported in the paper do not make the improvement clear.  The “local impact” metric reported in the paper does not seem like a good way of assessing improvement since (as the authors admit) it multi-counts improvements from one step in subsequent steps.  I suggest removing that metric for the more grounded global metric.  The charts reported in Figure 3 are not understandable or analyzed in the text.  Why are some of the sub-plots blank?  And where are the labels saying which chart is associated with each of the 4 variants?   We can’t tell which problem setting matches each chart.  Finally, while the green line (new method) outperforms the baseline in the majority of charts, there are some where it does not.  But no analysis or explanations are provided despite there being plenty of room in the paper.

Finally on the empirical side, the authors reference the work of Shen et al. who applied DQN to this ITS problem, but the authors do not provide an empirical comparison to this DQN based approach.  Since that is the state of the art, it seems necessary to apply that algorithm here and see if the gains over the baseline are comparable to the new algorithm.

On the algorithm side, while most of the approach is fairly clear, the description of the “Type 1 agent” omits crucial details about how items or subject areas are actually selected.  Unlike the other 3 cases, where actions are clearly related to items, Type 1 has an action that moves a threshold.  But how is that helpful?  Can’t an agent just move the threshold very low so it thinks all students have mastered all skills?  And how does moving a threshold determine an item to be given to a student.  More detail is needed to understand this case, which seems much different from the other 3.

On terminology, the paper often uses terms (such as (Guess, Slip, Learn…). on page 3 or  “b” in the last paragraph of page 3)or presents results (for instance the thetas in table 1) before the definitions of these terms.  Since readers at ICLR are unlikely to have an IRT background, the definitions on page 4 need to be moved up to a terminology section towards the beginning.
On related work, the paper did a good job referencing several very recent papers but failed to reference some of the papers related to core concepts and also lacks references to other slightly older works that used the same mix of student knowledge tracing with offline RL.

Examples:
HOT-DINA first appears on page 3 but no citation is given
Item response theory is mentioned on page 3 with no citation

Other RL for tutoring systems work;
“Cognitive modeling to represent growth (learning) using Markov decision processes” – builds a Bayesian representation of student skills and then uses a POMDP to plan in the belief space over skills (similar to the current work’s representation)

“Learning a Skill-Teaching Curriculum with Dynamic Bayes Nets” – similar to current work, calibrates a Bayes Net based on student data  and uses RL to create new policies.

The last sentence of the first paragraph of section 3.1 seems to be an author’s note or question to co-authors.

---

> ### Author Response · Authors · 2020-11-13
> **Thanks for helpful reviews**
>
> AnonReviewer2 - thanks a lot for your valuable reviews/feedback. We shall keep these comments in mind as we try to improve our work in the future.

---

### Official Review · AnonReviewer1 · 2020-10-28
**Paper is not about representation learning and does not motivate differences from related work**

**Rating:** 2
**Confidence:** 5

**Review:**

The paper intends to contribute “a novel framework for optimizing instructional sequencing in ain intelligent tutoring system”. More specifically, this framework uses deep reinforcement learning and evaluates learned policies on historical data.

A major strength of this paper is working with real human data from an application with obvious positive human impact. That working with this rich data comes necessarily with only working with a small amount of data is understandable, and it is not a weakness of the paper.

The most significant weakness of the paper is that it does not articulate a contribution that centers on representation learning -- the focus of this conference. A representation of student knowledge over time is learned (the 118-parameter HOT-DINA model), but this representation was already contributed by past researchers and does not seem to match the intended contribution of the paper. An action policy for controlling tutor behavior as a function of student knowledge state is also learned, but policy (or its internal details) are not examined from a representation learning standpoint. (For example, does some aspect of the different learned student-specific policies appear to recover/mimic another known aspect of those student identities? If so, would that be a good or bad result?)

The next most significant weakness is that when the paper makes novel choices, those choices are neither evaluated nor strongly motivated based on the past literature. Why PPO over DQN? (PPO can work in certain settings that DQN cannot, but past work already demonstrated DQN for a very similar setting.) Why HOT-DINA over BKT? (HOT-DINA is a much more expressive model, but the small amount of valuable historical data for this setting may limit the effectiveness of expressive models due to overfitting.)

Additional weaknesses are noted in the additional section-by-section feedback below.

This reviewer recommends (2) strong rejection. This is not inherently a paper about representation learning, and, even as a generic applied machine learning paper, it does not sufficiently motivate or evaluate the intended contribution of “a novel framework” for using machine learning in a specific application.

Questions for the authors:

* Are there structural reasons why Q learning (e.g. DQN) cannot be applied in this setting?
* Is there a way to verify that HOT-DINA is not overfitting in a way that makes evaluating the system on the same historical information used to train it unsound?

Section-by-section notes:

Title
- Focus on “instructional sequence policies” (for ITS, using RL), was hoping for something that sounded immediately relevant to representation learning.

Abstract
- It sounds like the method might not be as important as the application.
Hopefully we’ll hear more about the representation learned because this is an ICLR submission.

Introduction
- The introduction doesn’t state the intended conclusion or motivate the novel parts of the work for the reader.
- There’s a learned policy in here, but what is the learned representation you want to highlight for this specific venue on representation learning (ICLR)?

Simulating
- This is how you fit a pre-existing representation to new data, it doesn’t seem like this is the contribution of the paper.

Training
- Missing use of domain knowledge:
  * Is the the 100 timesteps considered in PPO training based on 100 being a typical number of interaction steps with the ITS among the Tanzanian students?
  * Even with a finite horizon, the rate at which students decide to exit the activity could be used to motivate a discount factor < 1. Presumably you have this information as well.
- Notes like “Useful to mention this?” suggest the paper was submitted in an incomplete state.
- It’s interesting to list the design alternatives for representing actions, but each should be contextualized with references to past research that used something similar.

Evaluating
- The evaluation in terms of local and global impact is unfamiliar to this reviewer (who knows other RL+ITS work). Not enough information is given to pin down exactly what local impact measures.
- What is the source and meaning of the historical baseline number?
- Figure 3 is too busy to interpret. Consider presenting it as a chart that aggregates across students using a single line (plus error bars) to represent the mean (plus stddev) state over time for the two policies.
- The way student-specific models are “evaluated against historical data” on those specific students suggests we are just testing on the training data. Why is this a valid methodology for this application? Cite past work on evaluating RL-based ITS systems to motivate your methods.

Relationship
- Consider covering work by others much much earlier in the paper so that the reader can understand why you made the choice you did (and that you can convince them that you know RL has been applied to ITS across many decades previously).
- Near “STEP uses a more powerful deep RL method” -- it is true that policy gradient can be applied in certain applications where Q-learning cannot, but if we aren’t given a note as to whether this is the case in the current application. Based on previous work, it seems like Q learning was applicable. Thus, using policy gradient methods (including PPO) would seem to add needless complication.
- This section indeed state how the current work is different from past work, but it does not motivate the differences. If others were successful with different methods, why change them for this paper?

Conclusion
- “This paper contributes a novel framework” -- what is novel is the combination of the HOT-DINA student knowledge model with the PPO reinforcement learning approach (within a larger framework shared by many other papers).

---

> ### Author Response · Authors · 2020-11-13
> **Thanks for helpful reviews**
>
> AnonReviewer1 - thanks a lot for your valuable reviews/feedback. We shall keep these comments in mind as we try to improve our work in the future.

---

### Decision · Program_Chairs · 2021-01-07
**Final Decision**

**Decision:**

Reject

**Comment:**

This paper introduces an important and interesting problem and a potentially interesting approach. Unfortunately, the reviewers agree that the current version isn't appropriate for ICLR in its current form. However, hopefully the feedback will be useful for the authors in revising and resubmitting this paper to another venue.